# Atypical intrinsic neural timescale in autism

Takamitsu Watanabe[1,2]*, Geraint Rees[1,3], Naoki Masuda[4]*

[1]Institute of Cognitive Neuroscience, University College London, London, United Kingdom; [2]RIKEN Centre for Brain Science, Wako, Japan; [3]Wellcome Trust Centre for Human Neuroimaging, University College London, London, United Kingdom; [4]Department of Engineering Mathematics, University of Bristol, Bristol, United Kingdom

**Abstract** How long neural information is stored in a local brain area reflects functions of that region and is often estimated by the magnitude of the autocorrelation of intrinsic neural signals in the area. Here, we investigated such intrinsic neural timescales in high-functioning adults with autism and examined whether local brain dynamics reflected their atypical behaviours. By analysing resting-state fMRI data, we identified shorter neural timescales in the sensory/visual cortices and a longer timescale in the right caudate in autism. The shorter intrinsic timescales in the sensory/visual areas were correlated with the severity of autism, whereas the longer timescale in the caudate was associated with cognitive rigidity. These observations were confirmed from neurodevelopmental perspectives and replicated in two independent cross-sectional datasets. Moreover, the intrinsic timescale was correlated with local grey matter volume. This study shows that functional and structural atypicality in local brain areas is linked to higher-order cognitive symptoms in autism.

DOI: https://doi.org/10.7554/eLife.42256.001

*For correspondence:
takamitsu.watanabe@ucl.ac.uk (TW);
naoki.masuda@bristol.ac.uk (NM)

**Competing interests:** The authors declare that no competing interests exist.

## Introduction

How long neural information is likely to be stored in a neural area is a fundamental functional property of the local brain region (*Chen et al., 2015*; *Hasson et al., 2015*; *Himberger et al., 2018*), and has been quantified as temporal receptive window (*Hasson et al., 2008*; *Chaudhuri et al., 2015*; *Honey et al., 2012*; *Lerner et al., 2011*; *Stephens et al., 2013*; *Yeshurun et al., 2017*), temporal receptive field (*Cavanagh et al., 2016*), or intrinsic neural timescale (*Murray et al., 2014*; *Kiebel et al., 2008*; *Gollo et al., 2015*; *Cocchi et al., 2016*). Computational studies propose that such neural timescales should show a rostrocaudal gradient in the brains (*Kiebel et al., 2008*) and that densely interconnected central regions, such as prefrontal and parietal areas, should have slower timescales compared to peripheral sensory areas (*Chaudhuri et al., 2015*; *Gollo et al., 2015*). These proposals are supported by empirical observations. Human neuroimaging and macaque electrophysiology studies show that neural timescales tend to be longer in frontal and parietal areas compared to sensory-related regions (*Honey et al., 2012*; *Stephens et al., 2013*; *Murray et al., 2014*; *Ogawa and Komatsu, 2010*), and suggested that such a prolonged neural timescale enables these higher-order brain cortices to integrate diverse information for robust sensory perception (*Hasson et al., 2008*; *Lerner et al., 2011*; *Stephens et al., 2013*; *Yeshurun et al., 2017*; *Ogawa and Komatsu, 2010*; *Gauthier et al., 2012*), stable memory processing (*Hasson et al., 2015*; *Murray et al., 2014*; *Bernacchia et al., 2011*), and accurate decision making (*Cavanagh et al., 2016*; *Runyan et al., 2017*). A recent brain stimulation study directly demonstrates that such a hierarchy of the intrinsic timescale is closely related to functional interactions between lower and higher brain regions (*Cocchi et al., 2016*). The heterogeneity of the neural timescale is considered to be a basis of the functional hierarchy in the brain (*Chen et al., 2015*; *Hasson et al.,*

**eLife digest** Autism is a brain disorder that affects how people interact with others. It occupies a spectrum, with severe autism at one end and high-functioning autism at the other. People with severe autism usually have intellectual impairments and little spoken language. Those with high-functioning autism have average or above average IQ, but struggle with more subtle aspects of communication, such as body language. As well as social difficulties, many individuals with autism show repetitive behaviors and have narrow interests.

The brains of people with autism process information differently to those of people without autism. The brain as a whole shows less coordinated activity in autism, for example. But whether individual brain regions themselves also work differently in autism is unclear. Watanabe et al. set out to answer this question by using a brain scanner to compare the resting brain activity of high-functioning people with autism to that of people without autism.

In both groups, networks of brain regions increased and decreased their activity in predictable patterns. But in individuals with autism, sensory areas of the brain showed more random activity than in individuals without autism. The most random activity occurred in those with the most severe autism. This suggests that the brains of people with autism cannot hold onto and process sensory input for as long as those of neurotypical people. By contrast, a brain region called the caudate showed the opposite pattern, being more predictable in individuals with autism. The most predictable caudate activity occurred in those individuals with the most inflexible, repetitive behaviors. These differences in this neural randomness appear to result from changes in the structure of the individual brain regions.

The findings of Watanabe et al. suggest that changes in the structure and activity of small brain regions give rise to complex symptoms in autism. If these differences also exist in young children, they could help doctors diagnose autism earlier. Future studies should investigate whether the differences in brain activity cause the symptoms of autism. If so, it may be possible to treat the symptoms by changing brain activity, for example, by applying magnetic stimulation to the scalp.
DOI: https://doi.org/10.7554/eLife.42256.002

*2015*; *Himberger et al., 2018*; *Chaudhuri et al., 2015*; *Gollo et al., 2015*; *Cocchi et al., 2016*; *Kukushkin and Carew, 2017*; *Friston and Kiebel, 2009*).

Given such fundamental roles of local neural dynamics in highly-organised information processing in the brain, we hypothesised that atypical intrinsic neural timescales should be observed in autism. In fact, the core symptoms of this prevalent neurodevelopmental disorder — challenges in socio-communicational skills and repetitive, restricted behaviours (RRB) — are often linked to atypical information processing (*Happé and Frith, 2006*; *Palmer et al., 2017*; *Booth and Happé, 2018*): weak coherence theory suggests that autism spectral disorder (ASD) is associated with impairments of the global integration of diverse information and over-enhancement of individual inputs (*Happé and Frith, 2006*; *Booth and Happé, 2018*); a recent Bayesian view also attributes autism to overweighing of local sensory information (*Palmer et al., 2017*; *Lawson et al., 2017*). These theories suggest that measures of local neural dynamics — such as intrinsic neural timescales — should be linked to the symptomatology of individuals with ASD.

Despite such theoretical implications, no study has investigated intrinsic timescales of neural signals in autism. Here, we aimed at exploring this local neural property in the brains of high-functioning individuals with ASD and examining its associations with the core symptoms of this condition.

## Results

First, we introduced a measurement of the intrinsic neural timescales for resting-state fMRI (rsfMRI) signals, and validated it using simultaneous EEG-fMRI data (*Deligianni et al., 2016*; *Deligianni et al., 2014*). We then applied the index to a rsfMRI dataset recorded from high-functioning adults with ASD and its demographically-matched controls, and searched for brain regions whose atypical neural timescales were associated with the ASD symptoms. Next, using a longitudinal rsfMRI dataset collected from adolescent children, we examined developmental trajectories of the local neural dynamics of these regions. Finally, we explored neuroanatomical bases of the intrinsic

neural timescales. The reproducibility of our findings was tested using two independent MRI datasets.

## Timescales of resting-state fMRI signals

Based on previous macaque studies (*Chaudhuri et al., 2015*; *Cavanagh et al., 2016*; *Murray et al., 2014*; *Bernacchia et al., 2011*), we calculated an intrinsic neural timescale by assessing the magnitude of autocorrelation of the resting-state brain activity. First, we estimated the sum of autocorrelation function (ACF) values in the initial positive period of the ACF (i.e., the sum of the area of the green bars in *Figure 1a*). The upper limit of this period was set at the discrete time lag value just before the one where the ACF became non-positive for the first time. To adjust for differences in the temporal resolution of the neural data, we then multiplied the obtained sum of ACF values by the repetition time (TR) of the fMRI recording. This product was used as an index for intrinsic neural timescales.

This definition was validated by comparing the fMRI-based timescale index to that based on neural data with a higher temporal resolution (here, simultaneously recorded EEG data (*Deligianni et al., 2016*; *Deligianni et al., 2014*); *Figure 1—figure supplement 1*). The fMRI-based timescales were strongly correlated with those based on the gamma-band EEG signals (adjusted $R^2$ = 0.71; *Figure 1b*; see *Figure 1—figure supplement 2* for other EEG bands). In addition, when the EEG signals were convolved with the hemodynamic response function (HRF), the intrinsic timescales based on the HRF-convolved EEG signals became closer to those based on fMRI signals (adjusted $R^2$ = 0.61; *Figure 1c*).

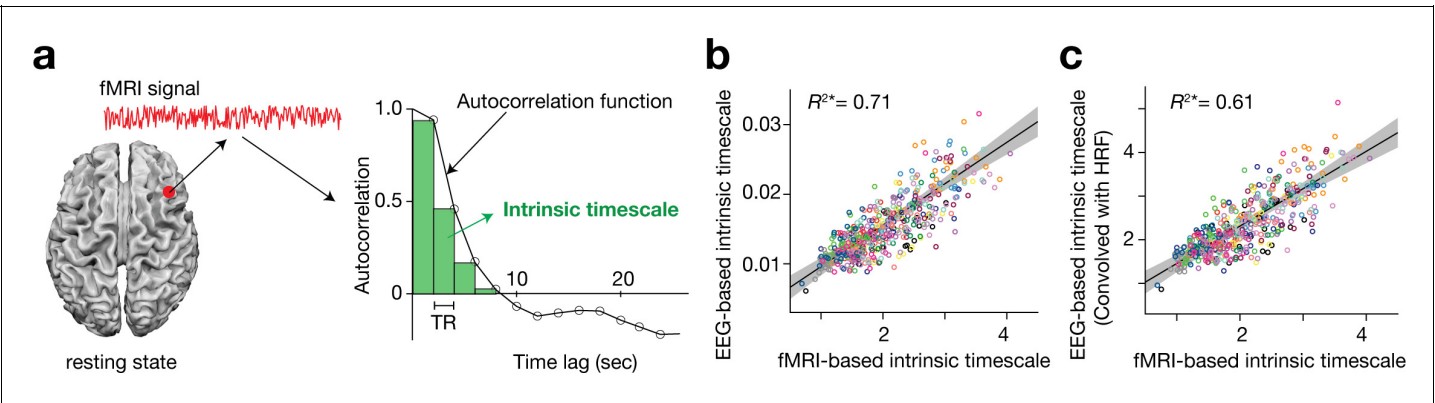

**Figure 1.** Definition and validation of intrinsic timescales based on resting-state fMRI data. (**a**) To estimate an intrinsic neural timescale of an fMRI signal, we first calculated the sum of autocorrelation function (ACF) values of the signals in the initial positive period of the ACF. The period is the area under the ACF up to the time lag value just before the one where the ACF becomes non-positive for the first time as the time lag increases. We then multiplied the obtained area under the ACF by the repetition time (TR), which defined the index for the intrinsic timescale. Open circles in the right panel show the empirical ACF values for the resting-state fMRI data. TR was 2 s in the current dataset. (**b and c**) fMRI-based intrinsic timescale scores were highly correlated with those calculated from simultaneously recorded EEG data (gamma band, adjusted $R^2$ = 0.71; panel b; see *Figure 1—figure supplement 2* for other EEG bands). The fMRI-based timescales were different from those based on the EEG data by two orders of magnitude. When we convolved the EEG signals with the hemodynamic response function (HRF), the intrinsic timescales based on EEG data showed the same magnitude as those based on fMRI data (panel c). A circle represents a combination of a brain region and a participant. The fMRI-based intrinsic timescale represents the index value averaged over a 4mm-radius sphere whose centre was determined by source reconstruction of independent components of EEG data. Different colours indicate data recorded from different participants. The grey area indicates 95% confidence interval.

DOI: https://doi.org/10.7554/eLife.42256.003

The following figure supplements are available for figure 1:

**Figure supplement 1.** Power spectrum of the preprocessed EEG data.

DOI: https://doi.org/10.7554/eLife.42256.004

**Figure supplement 2.** Comparisons between fMRI-based and EEG-based intrinsic timescales.

DOI: https://doi.org/10.7554/eLife.42256.005

## Intrinsic neural timescales in adults with ASD

Based on this formulation, we compared the intrinsic neural timescale between 25 high-functioning adults with ASD and 26 age-/sex-/IQ-matched typically developing (TD) individuals (*Table 1*) (*Di Martino et al., 2014*).

Both ASD and TD groups showed a similar whole-brain pattern of intrinsic neural timescales: longer timescales in frontal and parietal cortices and shorter timescales in sensorimotor, visual, and auditory areas (*Figure 2a*). This observation is consistent with previous reports about a hierarchal topography of timescales of local neural activity in brains of mice (*Runyan et al., 2017*), monkeys (*Chaudhuri et al., 2015*; *Murray et al., 2014*; *Ogawa and Komatsu, 2010*), and humans (*Hasson et al., 2008*; *Honey et al., 2012*; *Lerner et al., 2011*; *Stephens et al., 2013*; *Yeshurun et al., 2017*).

However, we also identified significant differences between the two groups (*Table 2*; $P_{FDR}$ <0.05). Individuals with ASD had a significantly shorter intrinsic timescale than TD individuals in bilateral postcentral gyri, right inferior parietal lobule (IPL), right middle insula, bilateral middle temporal gyri (MTG), and right inferior occipital gyrus (IOG) (*Figure 2b and d*), whereas the intrinsic timescale in the right caudate was significantly larger in the ASD group (*Figure 2c and e*).

## Associations between intrinsic timescales and core symptoms of ASD

We then tested for any associations between the observed atypical intrinsic neural timescales and the severity of autism, as measured by the Autism Diagnostic Observation Schedule (ADOS) (*Lord et al., 1989*). Because previous studies indicate that atypical neural information processing is a common basis for various ASD symptoms (*Happé and Frith, 2006*; *Belmonte et al., 2004*; *Watanabe and Rees, 2017*), we first examined associations between the neural timescales and the overall severity of this disorder (ADOS total scores). When no significant link was found in this analysis, we then calculated associations between the neural timescales and specific core symptoms.

Of the seven brain regions of interest (ROIs) whose intrinsic timescales were shorter in the ASD group (*Figure 2b and d*, *Table 2*), the bilateral postcentral gyri and right IOG showed negative correlations between the intrinsic timescale and overall severity of autism ($rho \leq -0.49$, $P_{uncorrected}$ <0.01, $P_{FDR}$ <0.05; *Figure 3a*).

The right caudate, a single ROI whose intrinsic timescale was significantly longer in the ASD group (*Figure 2e*), did not show such an association with the overall ADOS score ($rho = 0.19$, p=0.34). However, its intrinsic timescale was longer in individuals with more severe repetitive, restricted behaviours (RRB), as measured by ADOS RRB scores ($F_{3,21} = 9.9$, p<0.001, main effect of ADOS RRB in a one-way ANOVA; Spearman's $rho = 0.57$, p=0.002; *Figure 3b*).

These brain-symptom associations were preserved even when we conducted this association analysis in a more statistically rigorous manner (*Figure 3—figure supplement 1*). That is, we applied the

**Table 1.** Demographic data.

| | Typically developing (TD) | Autism spectrum disorder (ASD) | *P* value |
|---|---|---|---|
| Number of participants | 26 | 25 | - |
| Age | 25.3 ± 6.3 (18.1–39.4) | 27.3 ± 7.9 (18.4–50) | 0.4 |
| Sex | Male | Male | - |
| Laterality | Right-handed | Right-handed | - |
| Full IQ | 112.6 ± 12.0 (89–131) | 109.4 ± 13.6 (90–132) | 0.4 |
| Verbal IQ | 112.1 ± 12.1 (88–130) | 106.6 ± 13.8 (83–130) | 0.2 |
| Performance IQ | 110.3 ± 10.4 (90–129) | 110.9 ± 15.9 (83–133) | 0.9 |
| ADOS Social | - | 4.3 ± 1.4 (1–8) | - |
| ADOS Communication | - | 7.6 ± 2.3 (4–11_ | - |
| ADOS RRB | - | 1.1 ± 1.2 (0–3) | - |
| Mean head motion (mm) | 1.1 ± 0.6 (0.21–2.4) | 1.5 ± 0.8 (0.25–2.5) | 0.1 |

Mean ±SD (min–max)

DOI: https://doi.org/10.7554/eLife.42256.006

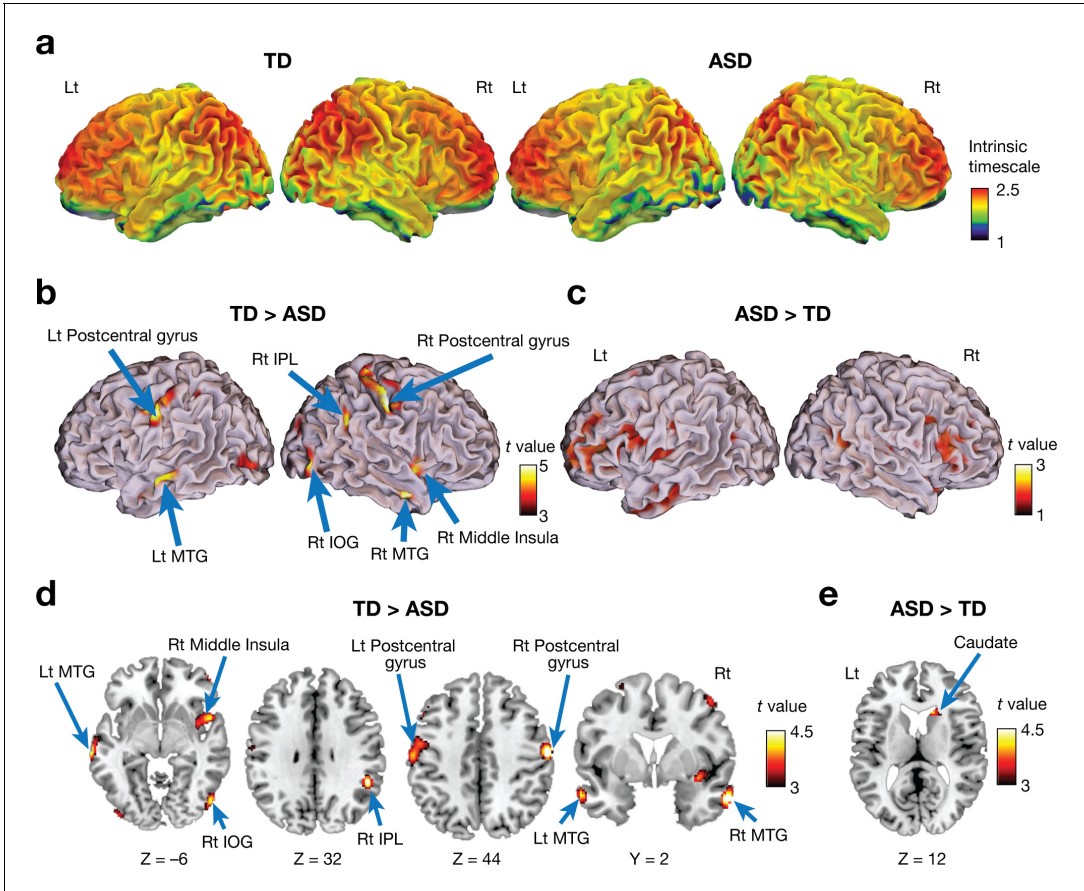

**Figure 2.** Voxel-wise comparison of the intrinsic neural timescale. (a) In both ASD and TD groups, we found longer intrinsic timescales in frontoparietal areas and shorter timescales in sensory-related areas. The colour bar indicates the intrinsic neural timescales (*Figure 1a*). (b–e) Individuals with ASD had significantly shorter intrinsic timescales in bilateral postcentral gyri, right inferior parietal lobule (IPL), right middle insula, bilateral middle temporal gyri (MTG), and right inferior occipital gyrus (IOG) (panels **b** and **d**), whereas the intrinsic timescale in the right caudate was significantly longer in the ASD group compared to the TD group (panels **c** and **e**).

DOI: https://doi.org/10.7554/eLife.42256.007

**Table 2.** Results of whole-brain intrinsic timescale analysis

|  |  | Coordinates |  |  |  |  |
| --- | --- | --- | --- | --- | --- | --- |
| Right/Left | Anatomical label | X | Y | Z | Cluster size | T value |
| TD > ASD |  |  |  |  |  |  |
| Right | Post-central gyrus | 58 | −14 | 44 | 470 | 5.2 |
| Left | Post-central gyrus | −58 | −14 | 40 | 309 | 4.3 |
| Right | Middle temporal gyrus | 60 | 2 | −26 | 170 | 4.8 |
| Left | Middle temporal gyrus | −70 | −26 | −6 | 321 | 4.3 |
| Right | Inferior occipital gyrus | 52 | −74 | −6 | 168 | 4.2 |
| Right | Inferior parietal lobule | 50 | −44 | 32 | 121 | 4.7 |
| Right | Middle insula | 50 | 10 | −4 | 228 | 4.3 |
| ASD > TD |  |  |  |  |  |  |
| Right | Caudate | 14 | 20 | 12 | 41 | 3.7 |

Threshold: $P_{FDR} < 0.05$.

DOI: https://doi.org/10.7554/eLife.42256.008

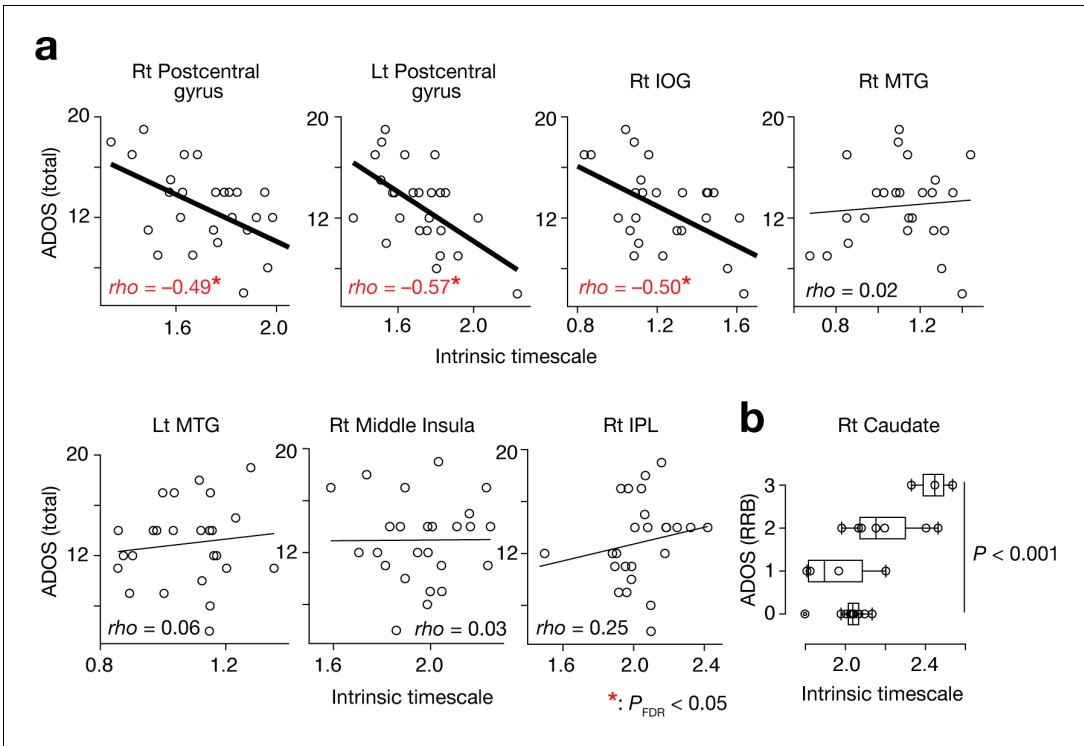

**Figure 3.** Associations between the intrinsic neural timescale and ASD symptoms. (a) We found negative associations between the intrinsic timescale and the overall severity of ASD in the bilateral postcentral gyri and right IOG but not in the bilateral middle temporal gyri (MTGs), right middle insula, and right inferior parietal lobule (IPL). The intrinsic timescales of the regions represent the averages of the timescale values within the clusters. The statistical threshold for the seven brain-behaviour comparisons was corrected by FDR. (b) The intrinsic neural timescale in the right caudate was not correlated with the overall severity of autism, but was associated with that of repetitive, restricted behaviours (RRB) of ASD.

DOI: https://doi.org/10.7554/eLife.42256.009

The following figure supplement is available for figure 3:

**Figure supplement 1.** Brain-symptom associations in independent datasets.

DOI: https://doi.org/10.7554/eLife.42256.010

same ROI sets to two independent fMRI datasets (ETH Zürich and Indiana University datasets; Supplementary Table 1 in *Supplementary file 1*) that were not used in the ROI search, and found negative correlations between the intrinsic timescales and the ADOS total scores in the bilateral postcentral gyri and right IOG ($rho \leq -0.60$) and a significant association between the intrinsic timescale and the ADOS RRB scores in the right caudate ($F \geq 6.0$, p$\leq$0.03 in one-way ANOVAs).

## Development of the intrinsic neural timescale in autism

We then examined whether these observations from adults with ASD could be seen in children with ASD. To this end, we analysed a longitudinal fMRI dataset recorded from adolescent children (two MRI scans for each participant, interval of the two scans = $2.8 \pm 0.4$ years for ASD children, $3.0 \pm 0.4$ years for TD children; Supplementary Table 2 in *Supplementary file 1*) (*Di Martino et al., 2014*). We traced the developmental trajectories of the intrinsic neural timescales of the four ROIs whose intrinsic timescales were atypical in the ASD group and associated with the severity of the symptoms. These four ROIs were defined as clusters found in the whole-brain analysis using the adult fMRI data (*Figure 2*, *Table 2*).

In adolescence, we found that the intrinsic neural timescale in bilateral postcentral gyri and right IOG was consistently shorter in individuals with ASD compared to TD individuals ($F_{1,31} > 9.0$, $P_{uncorrected} < 0.005$, $P_{FDR} < 0.05$, main effect of diagnosis in repeated-measures two-way ANOVAs with a diagnosis [ASD/TD] × scan order [1 st/2nd] structure; *Figure 4a*). In addition, the decreases in the

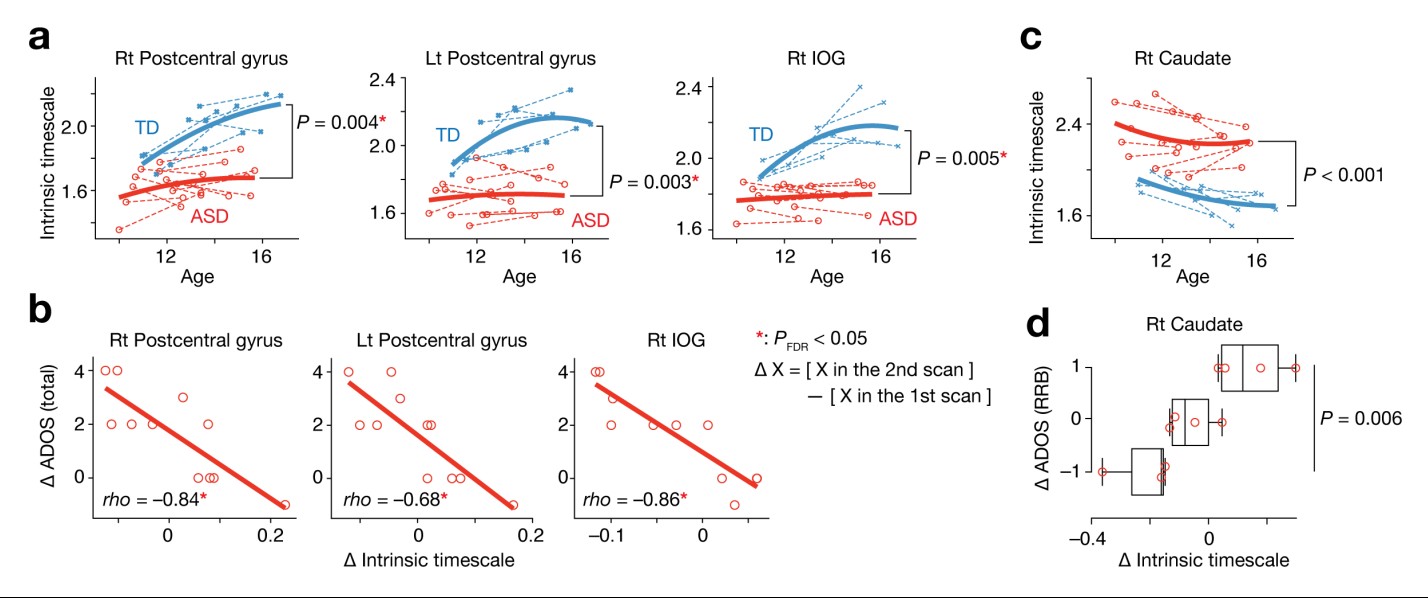

**Figure 4.** Analysis of a longitudinal developmental dataset. Atypical intrinsic neural timescales and their associations with symptoms of ASD were investigated in a longitudinal fMRI dataset obtained from adolescent children (Supplementary Table 2 in *Supplementary file 1*) (*Di Martino et al., 2014*). (a) In adolescence, the intrinsic neural timescales in the bilateral postcentral gyri and right inferior occipital gyri (IOG) were consistently shorter in children with ASD. Each bold curve indicates the quadratic curve fitted to individual data points for each group. Each dotted line represents a longitudinal change in each participant. (b) The underdevelopment of the intrinsic timescale was correlated with progression of overall ASD symptoms. The changes in the intrinsic timescale and ADOS score were defined as subtraction of these indices at the first scan from those at the second scan. (c) The intrinsic neural timescale in the right caudate was consistently longer in ASD group during adolescence. (d) The overdevelopment of the intrinsic timescale was correlated with progression of RRB symptoms.

DOI: https://doi.org/10.7554/eLife.42256.011

intrinsic timescale in these areas during this period were predictive of the increases in the overall severity of autism ($rho \leq -0.68$, $P_{uncorrected} <0.003$, $P_{FDR} <0.05$; *Figure 4b*).

In contrast, the intrinsic timescale in the right caudate was consistently longer in the group with ASD during adolescence ($F_{1,31} = 18.2$, $P_{uncorrected} <0.001$, main effect of diagnosis in a repeated-measures two-way ANOVA; *Figure 4c*), and the increase in the intrinsic timescale in this region was associated with progression of RRB symptoms ($F_{2,8} = 10.3$, p=0.006, main effect of ADOS RRB changes in a one-way ANOVA; *Figure 4d*).

These longitudinal observations are consistent with the cross-sectional findings (*Figures 2* and *3*) and suggest that autistic atypicality of temporal neural processing in local brain areas may already occur before adolescence.

## Associations between intrinsic neural timescale and local grey matter volume

Finally, we explored possible neuroanatomical bases for (or consequences of) intrinsic neural timescales by examining relationship with local grey matter volumes (GMVs). We focused on GMV because theoretically, an increase in neuronal density, which is measured by GMV (*Kanai and Rees, 2011*), would enhance recurrent neural network activity, and then enlarge the autocorrelation strength in the neural signals.

This theoretical assumption was validated by comparisons between intrinsic timescales and GMV across 360 brain areas (*Glasser et al., 2016*): at a group level, these functional and anatomical properties were positively correlated with each other (TD: $r = 0.40$, ASD: $r = 0.38$, p<$10^{-5}$; *Figure 5a*). Furthermore, the significant correlations were robustly observed at a single-participant level as well (TD: $r \geq 0.29$, ASD: $r \geq 0.28$, p<$10^{-5}$; *Figure 5b*). This association was also seen in the four brain regions whose atypical intrinsic neural timescale was associated with symptoms of autism ($r > 0.52$, p≤0.005; *Figure 5c*).

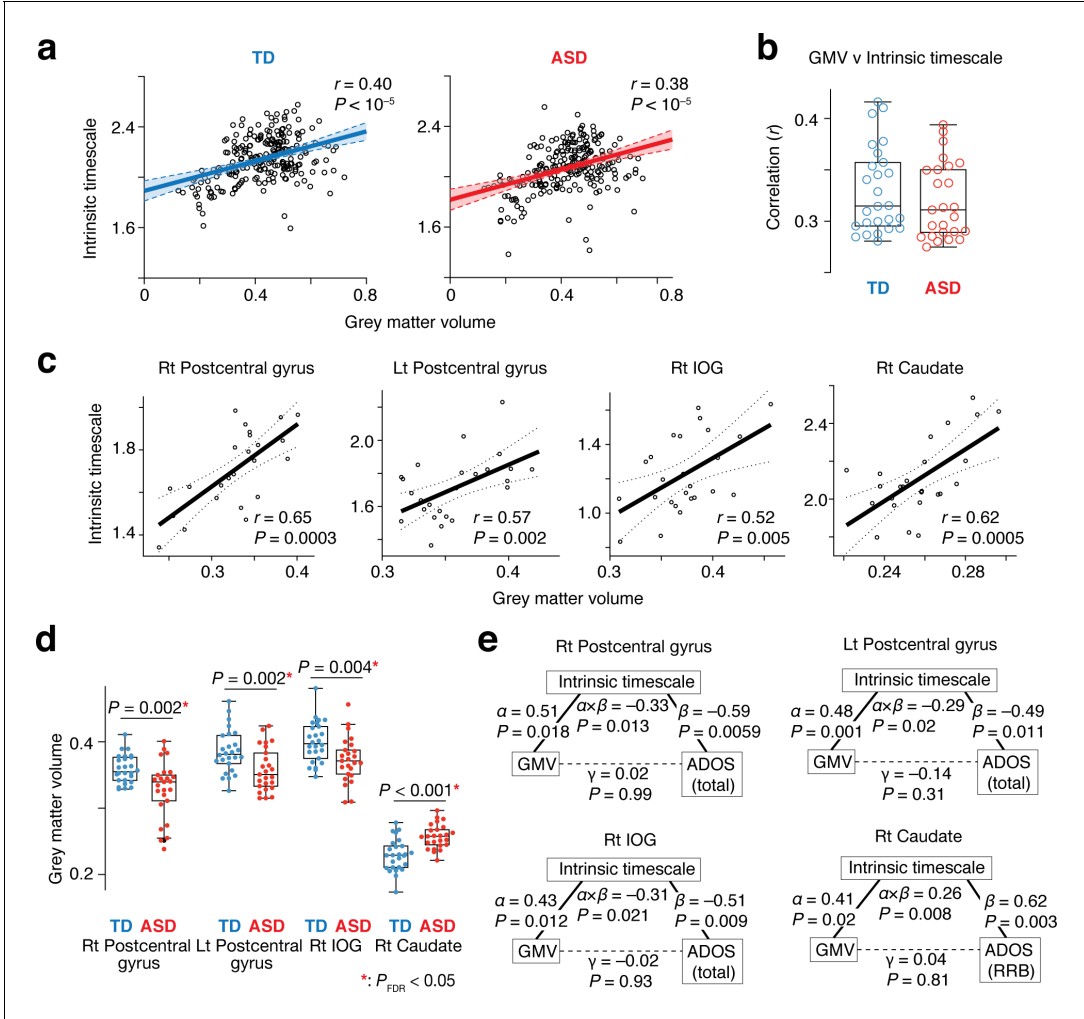

**Figure 5.** Neuroanatomical basis of the intrinsic neural timescale. (**a**) At a group level, intrinsic timescale was correlated with local grey matter volume (GMV). Each circle represents an across-participant average of the intrinsic timescale and that of the GMV in one of the 360 brain regions defined by a previous multi-modal brain parcellation study (**Glasser et al., 2016**). Dotted lines and shaded areas indicate 95% confidence intervals of the fitted lines. (**b**) The correlation between the intrinsic timescale and GMV was significant even at a single-participant level. Each circle represents the Pearson's correlation coefficient between the intrinsic timescale and GMV within each participant, which is calculated based on 360 brain areas (**Glasser et al., 2016**). (**c**) The correlation between the intrinsic timescale and GMV was seen in the four brain regions whose atypical intrinsic timescale was correlated with the severity of autism. Each circle represents an individual with ASD. Dotted lines indicate 95% confidence intervals of the fitted lines. (**d**) The GMVs of the three brain regions showing atypical reduction in the intrinsic timescale in autism were significantly smaller in the ASD than the TD group. In contrast, the GMV of the right caudate, which had an atypically longer intrinsic timescale in autism, was larger in the ASD group. (**e**) We conducted mediation analyses to examine the hypothesis that atypicality in the intrinsic timescale is one of the mediators linking atypical GMV and symptoms of autism. The analyses used the GMV, intrinsic timescale, and ADOS score as an independent variable, mediator variable, and dependent variable, respectively. '$\alpha$' indicates effects of GMV on the intrinsic timescale, and '$\beta$' denotes effects of the intrinsic timescale on ADOS scores. '$\gamma$' represents direct effects of GMV on ADOS scores, and '$\alpha \times \beta$' indicates indirect effects. The statistical significance of the indirect effects (i.e., $P$ values for '$\alpha \times \beta$') and the insignificance of the GMV-ADOS direct effects (i.e., $P$ values for '$\gamma$') support the working hypothesis.

DOI: https://doi.org/10.7554/eLife.42256.012

The following figure supplements are available for figure 5:

**Figure supplement 1.** Reproducibility test 1: ETH Zürich dataset.
DOI: https://doi.org/10.7554/eLife.42256.013
**Figure supplement 2.** Reproducibility test 2: Indiana University dataset.
DOI: https://doi.org/10.7554/eLife.42256.014

Given these findings together with atypical GMVs in the four brain regions in individuals with ASD ($t_{49}$ >3.0, $P_{uncorrected}$ ≤0.004 in two-sample $t$-tests, $P_{FDR}$ <0.05; *Figure 5d*), we can infer that intrinsic neural timescale is a mediator linking atypical GMV and ASD symptoms. This inference was, in fact, consistent with results of mediation analyses (*Figure 5e*).

## Reproducibility

We replicated our findings in the two independent MRI datasets obtained from adults with ASD, which were collected in ETH Zürich and Indiana University (Supplementary Table 1 in *Supplementary file 1*) (*Di Martino et al., 2014*).

In both datasets, ASD group yielded atypically shorter intrinsic timescales in the bilateral postcentral gyri and right IOG ($t \geq 4.0$, $P_{FDR}$ <0.05; *Figure 5—figure supplements 1a* and *2a*) and a longer intrinsic timescale in the right caudate ($t = 3.9$, $P_{FDR}$ <0.05; *Figure 5—figure supplements 1b* and *2b*). The shorter intrinsic timescale in the bilateral postcentral gyri and right IOG was correlated with the overall severity of ASD ($rho \leq$ –0.51; *Figure 5—figure supplements 1c* and *2c*), whereas the longer intrinsic timescale in the caudate was associated with RRB symptoms (p<0.028 in one-way ANOVAs; Spearman's $rho \geq$0.74; *Figure 5—figure supplements 1d* and *2d*). Moreover, the correlations between the intrinsic timescale and GMV were also replicated ($r \geq 0.49$; *Figure 5—figure supplements 1e* and *2e*).

## Discussion

We investigated the intrinsic neural timescale, whose length is closely related to the functional hierarchy in the brain (*Hasson et al., 2015*; *Murray et al., 2014*; *Cocchi et al., 2016*), in high-functioning individuals with autism. By calculating the time-dependent magnitude of autocorrelation function in resting-state fMRI (rsfMRI) signals, we found that in adults with ASD, the intrinsic timescale was significantly shorter in the bilateral postcentral gyri and right inferior occipital gyrus, and longer in the right caudate. The shorter intrinsic timescale in these primary sensory/visual areas in autism was correlated with the overall severity of autism. The longer intrinsic timescale in the caudate in autism was associated with the severity of repetitive, restricted behaviours. Moreover, this temporal property in local neural signals was linked to local grey matter volumes (GMVs). These findings indicate the possibility that functional and structural properties in local brain areas could have a critical influence on higher-order cognitive symptoms in autism.

Furthermore, we investigated the validity of these observations on longitudinal neuroimaging data recorded from adolescents with ASD. We found atypical development of the intrinsic timescale during adolescence in autism (*Figure 4a and c*) and identified significant associations between such atypical neural development and progression of ASD symptoms (*Figure 4b and d*). These findings imply that such atypicality in the temporal characteristics of local neural activity may be one of the basic neuro-aetiologies of autism, which needs to be tested using data collected from much younger children with ASD in future studies.

The association between the intrinsic timescales and GMVs is theoretically reasonable. Large GMVs are considered to indicate a high density of neurons in local brain regions (*Kanai and Rees, 2011*), which is thought to be accompanied with more synapses (*Cullen et al., 2010*) and greater synaptic weights (*Perin et al., 2011*). Moreover, computational studies suggest that such a high neuronal and synaptic density should increase reciprocal connections within the areas and enhance local clustering (*Perin et al., 2013*). Given that spontaneous neural activity largely depends on such recurrent neural networks (*Ikegaya et al., 2004*), resting-state neural activities of brain regions with large GMVs would show more repetition patterns and larger autocorrelations. Although future studies have to directly examine this hypothesis, this logic accounts for the significant correlations between the local neural dynamics and local neuroanatomical structures.

Some human neuroimaging researches examined autistic local neural dynamics in the sensory-related areas and reported observations that are consistent with the current findings. For example, fMRI studies that investigated intra-participant variability of brain signals across time found atypically large signal variability in the prefrontal region (*Dinstein et al., 2011*), somato-sensory area (*Haigh et al., 2015*; *Dinstein et al., 2012*), auditory area (*Haigh et al., 2015*; *Dinstein et al., 2012*), and primary visual cortex (*Dinstein et al., 2012*; *Dinstein et al., 2010*; *Milne, 2011*) in individuals with ASD. In particular, one study found that the severity of ASD was significantly correlated with

such atypical signal variability in the sensory/visual areas (*Dinstein et al., 2012*). If we can assume that such large variability of local brain signals indicates more random brain activity and consequently yields weak autocorrelations, these previous reports can be interpreted as being consistent with the current findings.

In contrast, local neural dynamics in the caudate in autism were poorly understood. In fact, the neuroanatomical association between the subcortical region and the RRB symptoms was reported in previous structural MRI studies (*Langen et al., 2014*; *Langen et al., 2009*; *Langen et al., 2007*; *Hollander et al., 2005*; *Schuetze et al., 2016*), which is consistent with the current observation about the caudate. However, to the best of our knowledge, no prior research has been conducted on intrinsic neural timescales or signal variability of the caudate in autism.

A recent review has suggested that the intrinsic timescale and TRW is not an artefact of neuroimaging signals but closely associated with an ability of local brain areas to pool, normalise, and complete information (*Himberger et al., 2018*). In particular, for sensory information processing, a longer neural timescale is considered to make brain responses more robust against fluctuations in sensory inputs and enable steady and consistent perception (*Himberger et al., 2018*; *Honey et al., 2012*; *Murray et al., 2014*). Given this, we speculate that the atypically short intrinsic timescale in the primary sensory/visual cortices observed in the ASD group (*Figures 2b* and *4a*) might potentially be one of neural bases of perceptual hyper-sensitivity often seen in autism (*American Psychiatric Association, 2013*).

This rsfMRI study did not adopt a formulation that has been used to measure neural timescales in previous human fMRI studies (*Hasson et al., 2015*; *Hasson et al., 2008*; *Lerner et al., 2011*; *Stephens et al., 2013*; *Yeshurun et al., 2017*; *Gauthier et al., 2012*), because this definition of neural timescale — so-called temporal receptive window (TRW) — was designed for task-related brain activity data and cannot be directly applied to resting-state fMRI data. Instead, based on previous non-human electrophysiology studies (*Cavanagh et al., 2016*; *Murray et al., 2014*; *Bernacchia et al., 2011*; *Runyan et al., 2017*), we defined the intrinsic neural timescale as the magnitude of autocorrelation of brain activity. In addition, to reduce adverse effects of the low sampling rates of fMRI recording, we did not conduct curve fitting to the autocorrelation coefficients as electrophysiological work did (*Cavanagh et al., 2016*; *Murray et al., 2014*; *Runyan et al., 2017*); we simply calculated the area under the autocorrelation function (ACF) to estimate the autocorrelation strength (*Figure 1a*).

We validated this definition of the intrinsic neural timescales using the simultaneously recorded EEG-fMRI data. Although the fMRI-based neural timescale was different from the EEG-based one by two orders of magnitude, the two measures were significantly correlated with each other (*Figure 1b* and *Figure 1—figure supplement 2*). Moreover, such a difference would be reasonable because an EEG signal is likely to peak ~100 ms after a stimulus onset and an fMRI signal — a product of neurovascular coupling (*Hillman, 2014*; *Martindale et al., 2003*) — tends to take 5 ~ 10 s to peak (*Logothetis et al., 2001*; *Yeşilyurt et al., 2008*). In fact, when we convolved the EEG signals with the hemodynamic response function (HRF) to take into account such neurovascular coupling, the resultant intrinsic timescales based on HRF-convolved EEG signals were similar in the magnitude to those based on the fMRI data (*Figure 1c*).

These EEG-fMRI comparisons indicate that the fMRI-based neural timescales represent an aspect of local neuronal activity. However, it is beyond the scope of this study to conclude that such an fMRI-based index reflects the same neuronal phenomena as those seen in previous electrophysiology work that calculated neural timescales from spike activity data (*Cavanagh et al., 2016*; *Murray et al., 2014*; *Ogawa and Komatsu, 2010*; *Runyan et al., 2017*). To clarify this issue, future studies have to directly compare the fMRI-based neural timescales with those based on neuronal spike activities that are collected simultaneously with fMRI data.

Our exploratory study identified significant associations between local brain dynamics and behavioural tendencies in autism; however, biological mechanisms underlying this brain-behaviour link remain unknown. Previous work proposed that such an atypical neural timescale may represent atypical functional hierarchy in information processing in brains (*Himberger et al., 2018*; *Chaudhuri et al., 2015*; *Gjorgjieva et al., 2016*). However, it is necessary to directly examine this hypothesis by analysing task-related whole-brain activity in ASD populations. Future work also needs to investigate mechanisms linking these local neural dynamics to atypical large-scale brain dynamics seen in autism (*Watanabe and Rees, 2017*).

Another limitation of this study is the relative homogeneity of the participants with ASD that we have studied. To improve detectability, we limited the ASD group to high-functioning right-handed adult males. Although we confirmed the main findings in an adolescent dataset (*Figure 4*), future studies have to examine the current observations in different subsets of ASD cohorts.

This resting-state fMRI study investigated how long local brain areas can store information in individuals with autism and identified a shorter intrinsic timescale in the bilateral primary sensory/visual cortices and a longer intrinsic timescale in the right caudate. This atypicality in local neural dynamics was associated with the severity of autism and also correlated with local grey matter volumes. Although these findings should be examined in larger and more diverse cohorts of individuals with ASD, our work highlights the importance of investigating neural dynamics in neuro-psychiatric disorders.

## Materials and methods

### Validation of the formulation of intrinsic neural timescale

We examined the validity of the current formulation of the intrinsic neural timescales by comparing simultaneously recorded resting-state EEG and fMRI data that were shared in the Open Science Framework (*Deligianni et al., 2016*; *Deligianni et al., 2014*).

### Data for the validation test

Simultaneous EEG-fMRI data were collected from 17 healthy adults (6 females, 32.84 ± 8.1 years old) at UCL under the ethical approval from the UCL Research Ethics Committee and informed consent obtained from all the participants (*Deligianni et al., 2016*; *Deligianni et al., 2014*). The data were obtained during rest, in which the participants were asked to open their eyes and remain awake with fixating a white cross on a black background.

### EEG data processing for the validation test

EEG data were recorded by an MRI-compatible EEG system with 64 channels (BrainCap MR, Germany) and were preprocessed in MATLAB (MathWorks, Inc) and EEGLAB (*Delorme and Makeig, 2004*) (sccn.ucsd.edu/eeglab/).

First, the EEG data were referenced to the average of all the electrodes, and downsampled to 250 Hz. After conducting band-pass filtering (1–80 Hz), an optimal basis set (OBS) algorithm based on principal component analysis (*Niazy et al., 2005*) was used to reduce the gradient artefacts induced by fMRI scanning. Cardio-ballistic artefacts (CBAs) were reduced as follows (*Jamison et al., 2015*; *Liu et al., 2012*): the alignment of the occipital CBAs was optimized with individual participant's heartbeat; then, EEG components that were strongly correlated with the occipital CBAs were identified and excluded by conducting independent component analysis (ICA) and calculating mutual information. The remaining EEG artefacts induced by eye blinks, eye movements, and muscle activity were removed by ICA. Next, we excluded epochs whose mean global field power was larger than five standard deviations above the mean across the entire recording. Finally, to identify the source location of these preprocessed EEG signals, we conducted source reconstructions of the remaining ICA-components using DIPFIT2 function implemented in EEGLAB, and obtained MNI coordinates for each of the independent components. To reduce ambiguity in the following fMRI analysis, we excluded independent components whose whole-brain activity patterns did not show clear laterality and thus whose sources were calculated to be in both brain hemispheres.

We confirmed that the remaining independent components of EEG data were effectively free from fMRI-oriented gradient noise. In fact, the power spectrums of the EEG data showed that the preprocessing procedures significantly reduced fMRI-induced noise in the EEG signals (*Figure 1—figure supplement 1a and b*). Therefore, we used these EEG data for the intrinsic timescale analysis.

We then filtered the preprocessed data to delta (1–4 Hz), theta (4–8 Hz), alpha (8–13 Hz), beta (13–30 Hz), and gamma (30–80 Hz) bands, and calculated a Hilbert envelope amplitude for each band wave (*Deligianni et al., 2014*). Using the envelope amplitudes, we estimated an intrinsic timescale in the same manner as used for resting-state fMRI signals (*Figure 1a*).

As for the gamma-band EEG signals, we convolved them with the hemodynamic response function (HRF) implemented in SPM12 and calculated the intrinsic timescales for the HRF-convolved EEG data (*Figure 1c*).

## MRI data processing for the validation test

The MRI data were collected in a 1.5T scanner (Avanto, Siemens) with a 12-channel head coil. Functional data were recorded using EPI sequence (TR 2.16 s, TE 30 ms, FA 75°, spatial resolution 3.3 mm cubic), and T1-weighted structural MRI data were also obtained (*Deligianni et al., 2016*; *Deligianni et al., 2014*). For each participant, these MRI data were preprocessed in the same manner as mentioned in the main text and calculated the intrinsic timescale for each voxel.

We then calculated the average intrinsic timescales for the brain areas corresponding to each independent component of the EEG data. We defined the brain areas as a 4mm-radius sphere whose centres were determined based on the MNI coordinates obtained in the source reconstructions of the EEG data. Through these analyses, we obtained fMRI-based intrinsic neural timescales and compared them with EEG-based ones using linear regression analyses.

## Participants for main analysis

This study used datasets shared in ABIDE (*Di Martino et al., 2014*). The main analysis was based on a dataset recorded from 25 high-functioning adults with autism spectrum disorder (ASD) and 26 typically developing (TD) controls in University of Utah (*Table 1*). We chose this dataset because of its largest size of high-functioning adults with ASD. We selected participants based on their age ($\geq$18 years old), sex (male), handedness (right-handed), IQ (full/verbal/performance IQ $\geq$ 80), and head motion during scanning (mean $\leq$3 mm). We focused on high-functioning right-handed male adults to reduce heterogeneity across individuals with ASD (*Jack and A Pelphrey, 2017*).

The diagnosis of ASD was made based on structured interviews by a clinical expert for ASD in accordance with ADOS and DSM-IV-TR (*Lord et al., 1989*). IQ was evaluated based on Wechsler Abbreviated Scale of Intelligence. Handedness was scored based on Edinburgh Handedness Inventory. The data collection was approved by the local ethics committees (University of Utah IRB), and all participants provided written consent.

## MRI data and preprocessing

Resting-state and anatomical MRI data were collected using a 3.0T MRI scanner (Magnetom Trio, Siemens; resting-state MRI, EPI sequence, TR 2 s, TE 28 ms, 40 slices, interleaved, FA 90°, 3.4 $\times$ 3.4 $\times$ 3.0 mm; anatomical MRI, T1-weighted sequence, TR 2.3 s, TE 2.91 ms, FA 9°, 1.0 $\times$ 1.0 $\times$ 1.2 mm). The resting-state MRI data were recorded for ~8 min for each participant, during which the participants were asked to relax with their eyes open.

The resting-state MRI data were preprocessed with SPM12 (www.fil.ucl.ac.uk/spm). After discarding the first five images, we performed realignment, unwarping, slice-timing correction, and normalisation to the standard template (ICBM 152). We then removed effects of head motion, white matter signals, and cerebrospinal fluid signals by regression analyses, and finally conducted band-pass temporal filtering (0.01–0.1 Hz). Note that we excluded participants whose mean head motions were more than 3 mm. After this exclusion, there was no significant difference in the mean head motion (p>0.1 in a two-sample *t*-test) and maximum/mean framewise displacement (FD) (maximum FD, p>0.2; mean FD, p>0.4 in a two-sample *t*-test) between the TD and ASD groups.

## Intrinsic neural timescale map

At a single-participant level, we used these preprocessed fMRI data to evaluate the intrinsic neural timescale for each voxel as follows. First, we estimated an autocorrelation function (ACF) of the fMRI signal of each voxel (time bin = TR), and then calculated the sum of ACF values in the initial period where the ACF showed positive values (i.e., the sum of the area of the green bars in *Figure 1a*). The upper limit of this period was set at the point where the ACF hits zero for the first time. After repeating this procedure for every voxel, we applied spatial smoothing to the brain map (Gaussian kernel, full-width at half maximum = 8 mm) to improve the signal-to-noise ratio. We used this whole-brain map as an intrinsic timescale map in which the value at each voxel is equal to the intrinsic neural timescale of the brain region.

After performing this calculation for all participants, we compared the intrinsic neural timescale maps between ASD and TD groups using a random-effects model. We searched for brain areas showing significant differences between the two groups ($P_{FDR}$ <0.05).

### Brain-symptom associations

For the brain regions whose intrinsic neural timescale was significantly different between the individuals with autism and the TD individuals (*Table 2*), we explored associations between their intrinsic neural timescales and the ASD symptoms. Because atypical information processing in autism could be a common basis for various ASD symptoms (*Happé and Frith, 2006*; *Belmonte et al., 2004*; *Watanabe and Rees, 2017*), we first calculated Spearman's correlation coefficients between the average intrinsic timescale in the regions and the overall severity of ASD (ADOS total score). The ADOS total scores were defined by the sum of the ADOS social, ADOS communication, and ADOS RRB scores. The regions of interest (ROIs) were defined as clusters found in the whole-brain analysis stated above (*Table 2*), and an intrinsic timescale for each ROI was given by the average within the corresponding cluster. The multiple comparisons between these brain regions were corrected by FDR.

When we found no significant correlations between the neural timescales and ADOS total scores, we calculated associations with the social and RRB symptoms, respectively. The social symptoms were measured as the sum of the ADOS social scores and ADOS communication scores. The associations with the RRB symptoms were evaluated in one-way ANOVA because the ADOS RRB scores were too sparse for an accurate correlation analysis.

### Confirmation of the brain-symptom associations

To minimise any statistical dependence between the brain-symptom association analysis and the ROI search, we repeated the association analysis by applying the same ROIs to two independent MRI datasets that were not used in the ROI search. The data were collected in ETH Zürich and Indiana University and shared through ABIDE (Supplementary Table 1 in *Supplementary file 1*) (*Di Martino et al., 2014*). The MRI data were collected under the approval of each local ethics committee in the recording site and with the written informed consent of all the participants.

Participants were selected based on the same criteria as those in the main analysis (age ≥18 years old, sex: male, handedness: right-handed, full/verbal/performance IQ ≥ 80, and mean head motion ≤3 mm). This selection excluded three ASD and nine TD individuals from the entire ETH Zürich dataset, and 11 ASD and 10 TD individuals from the Indiana University dataset. As a result, this reproducibility test analysed 10 ASD and 15 TD individuals for the ETH Zürich dataset and 9 ASD and 10 TD individuals for the Indiana University dataset.

After conducting the same preprocessing as in the original analysis, we calculated an intrinsic neural timescale for each voxel, and extracted intrinsic timescales for the four ROIs that were defined in the main analysis (Rt/Lt postcentral gyrus, Rt IOG, and Rt caudate). We then tested for the associations between the neural timescale at these ROIs and the severity of ASD.

### Analysis of longitudinal developmental data

We examined the main findings from the perspective of neurodevelopment. To this end, we analysed longitudinal MRI data that were collected from 11 high-functioning adolescent children with ASD and seven age-/sex-/IQ-matched TD children in University of California Los Angeles (two scans for each participant; Supplementary Table 2 in *Supplementary file 1*).

The MRI data were preprocessed in the same manner as in the main analysis, and an intrinsic timescale was estimated for each voxel at each time point in each participant. We then extracted intrinsic timescales for the eight brain regions of interest (ROIs). The ROIs were defined as clusters found in the main MRI analysis (*Table 2*), and the neural timescales of the regions were given by the average within the corresponding clusters.

For each ROI, we compared developmental trajectories of the intrinsic timescale between ASD and TD groups. In addition, we examined whether such developmental changes in the intrinsic timescale are related to changes in clinical severity of autism. We calculated the intrinsic timescale changes by subtracting the intrinsic timescale at the first scan from that at the second scan. The changes in ADOS scores were quantified in the same manner.

## Grey matter volume v intrinsic timescale

We investigated the neuroanatomical bases for intrinsic timescale by comparing grey matter volume (GMV) to the temporal property of neural signals. GMV was calculated from structural MRI data using SPM12 as follows: the MRI images were segmented into grey matter, white matter, and cerebrospinal fluid using the New Segment Toolbox (*Ashburner and Friston, 2005*); using the DARTEL Toolbox (*Ashburner, 2007*), the segmented grey matter images were aligned, warped to a template space, resampled to 1.5 mm isotropic voxels, and registered to a participant-specific template.

At a whole-brain level, we first segmented these preprocessed grey matter images into 360 areas according to a recently proposed multi-modal brain parcellation system (*Glasser et al., 2016*), and extracted GMV from each area. By applying the same parcellation system to the whole-brain map of the intrinsic timescale, we calculated average intrinsic timescale for each brain segment. We then averaged these anatomical and functional metrics across participants for each brain segment, which yielded a group-average GMV map and a group-average intrinsic timescale map for each group. By comparing these maps with linear regression analyses, we estimated associations between intrinsic timescale and GMV.

Next, we examined this function-anatomy correlation in the brain regions whose intrinsic timescale significantly deviated in autism and showed significant associations with the severity of ASD symptoms.

Finally, we performed mediation analyses to investigate correlations between intrinsic timescale, GMV, and severity of ASD after normalising these functional, anatomical, and clinical scores.

## Reproducibility tests

We examined reproducibility of the main findings with the two independent MRI datasets that were used in the confirmatory brain-symptom associations (see 'Confirmation of the brain-symptom associations' in this Materials and methods section; Supplementary Table 1 in *Supplementary file 1*). We repeated the same voxel-wise comparison of the intrinsic timescale between the ASD and TD groups, calculated associations between the intrinsic timescale and clinical scores in the ASD group, and assessed correlations with GMV.

## Acknowledgements

This work was supported by a Marie-Curie Individual Fellowship from European Commission (656161; TW), JSPS (18H06094), Yamaha Sports Challenge Fellowship (TW), Fukuhara Fund for Applied Psychoeducation Research (TW), SENSHIN Medical Research Foundation (TW), a Wellcome Trust Senior Clinical Research Fellowship (100227; GR), and JST CREST (JPMJCR1304; NM).

## Additional information

### Funding

| Funder | Grant reference number | Author |
| --- | --- | --- |
| European Commission | 656161 | Takamitsu Watanabe |
| Japan Society for the Promotion of Science | 18H06094 | Takamitsu Watanabe |
| Yamaha | Yamaha Sports Challenge Fellowship | Takamitsu Watanabe |
| Fukuhara Fund for Applied Psychoeducation Research | | Takamitsu Watanabe |
| SENSHIN Medical Research Foundation | | Takamitsu Watanabe |
| Wellcome Trust | 100227 | Geraint Rees |
| Japan Science and Technology Agency | JPMJCR1304 | Naoki Masuda |

The funders had no role in study design, data collection and interpretation, or the decision to submit the work for publication.

## Author contributions
Takamitsu Watanabe, Conceptualization, Data curation, Software, Formal analysis, Funding acquisition, Validation, Investigation, Visualization, Methodology, Writing—original draft, Project administration, Writing—review and editing; Geraint Rees, Supervision, Funding acquisition, Project administration, Writing—review and editing; Naoki Masuda, Conceptualization, Funding acquisition, Writing—review and editing

## Author ORCIDs
Takamitsu Watanabe (iD) http://orcid.org/0000-0002-8104-6873

## Ethics
Human subjects: The data collection was approved by the local ethics committees in the data recording sites, and all participants provided written consent.

## Decision letter and Author response
Decision letter https://doi.org/10.7554/eLife.42256.020
Author response https://doi.org/10.7554/eLife.42256.021

# Additional files

## Supplementary files
• Supplementary file 1. Supplementary table 1: Dataset for the reproducibility test. Supplementary table 2: Properties of the longitudinal dataset.
DOI: https://doi.org/10.7554/eLife.42256.015

• Transparent reporting form
DOI: https://doi.org/10.7554/eLife.42256.016

## Data availability
All the data used here were shared in ABIDE (http://fcon_1000.projects.nitrc.org/indi/abide/) or Open Science Framework (https://osf.io/94c5t/). From the ABIDE site, the data submitted by the University of Utah (ABIDE I), ETH Zürich (ABIDE II), Indiana University (ABIDE II), and UCLA (longitudinal, ABIDE II) were used. To access these datasets, users first need to register with the NITRC and 1000 Functional Connectomes Project (further information here http://fcon_1000.projects.nitrc.org/indi/req_access.html).

The following previously published dataset was used:

| Author(s) | Year | Dataset title | Dataset URL | Database and Identifier |
|---|---|---|---|---|
| Clayden J, Deligianni F | 2016 | Data from: NODDI and tensor-based microstructural indices as predictors of functional connectivity. | https://osf.io/94c5t/ | Open Science Framework, 94c5t |

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
