## [Decision Letter]

Thank you for submitting your article "Atypical intrinsic neural timescale in autism" for consideration by *eLife*. Your article has been reviewed by three peer reviewers, including Michael Breakspear as the Reviewing Editor and Reviewer #1, and the evaluation has been overseen by Joshua Gold as the Senior Editor. The following individuals involved in review of your submission have agreed to reveal their identity: Leonardo L. Gollo (Reviewer #2); Warren W Pettine (Reviewer #3).

The reviewers have discussed the reviews with one another and the Reviewing Editor has drafted this decision to help you prepare a revised submission.

Summary:

All reviewers found the paper to be well motivated, clearly written and of substantial interest. No major statistical or design issues were identified. While the reproducibility data sets were laudable, in general all data sets are small to modest in size.

Some great clarity of analysis should be provided and closer linkages to computational work on time scale hierarchies in the brain should be provided.

None of the reviewers ask for substantial new work that could not be undertaken within a reasonable period of time. All requests for revisions are straightforward and are provided from each of the reviewers below.

Reviewer #1:

Atypical intrinsic neural timescale in autism. I previously supplied some feedback to the authors on their original submission. It's a very interesting paper and has been appropriately revised.

1) The between group contrast, ASD>TD rests upon measures that are almost certainly used to estimate the severity of ASD: I'm therefore concerned there is some dependence between the test for finding the clusters (ASD>TD) and the regression against ASD severity – not quite double dipping, but surely some lack of complete independence. Please comment/respond to this statistical issue.

2) The developmental data set yields very intriguing findings: Are the ROI's used here those discovered in the first cohort? If so, this is important and should be highlighted. Either way, this is a small cohort and should be noted as such.

3) The authors provide nice correlations between fMRI and EEG-based intrinsic time scales (Figure 1): however these differ by two orders of magnitude: This difference should be highlighted in the Discussion and other potential factors, such as neurovascular, should be mentioned.

Reviewer #2:

Watanabe, Rees, and Masuda propose to compare the intrinsic timescales of brain regions in ASD and TD. They find significant differences in brain regions that are associated with the severity of symptoms and their gray-matter volume. This is an outstanding contribution to the understanding of structure, dynamics, and function in the human brain. The results are compelling and the analysis is thorough. I naturally recommend that the paper is published. To improve the presentation and the reproducibility of the results the authors should consider some minor suggestions:

1) The study finds a variety of regions that have atypical intrinsic timescales and the timescales of some of these regions have significant association with ADOS and with RRB. However, the authors do not discuss whether those regions have been reported before in other ASD studies. This should be discussed, and, if they have not been reported before, the authors might want to highlight the sensitivity of their new proposal.

Introduction

2) The authors mention "functional hierarchies" in the Introduction. Although this is a nice motivation, the concept of "hierarchy of timescales" should also be introduced. The hierarchy of timescales proposes a strong link between the well-established hierarchy in brain structure with a hierarchy in brain function with peripheral regions (at the bottom of the hierarchy) exhibiting fast timescales and core regions (at the top of the hierarchy) exhibiting slow timescales. This concept is proposed by Kiebel et al., 2008, and thoroughly explored by Gollo et al., 2015. Moreover, a distinct functional role of regions at the bottom and top of the hierarchy was demonstrated by Cocchi et al., 2016, using brain stimulation.

3) The authors mention "core symptoms" in the Introduction without providing additional details. A brief introduction and motivation of the different (social, communication, RRB) measures is missing in the manuscript, and would improve the presentation. Moreover, it is not clear how ADOS (total) was computed.

Results/Materials and methods

4) It is unclear why the results start with a negative sentence explaining what was not done. Instead of this approach, it would be better to provide a brief summary of what the results will cover (the cohorts, EEG, fMRI, structural imaging, symptom measures).

5) "The largest time lag in calculating the initial positive period was set to the value at which the ACF hit zero for the first time as the time lag was increased". This sentence is misleading because it suggests that an interpolation might be required to find when ACF hits zero, and it also contradicts the caption " The initial positive period is the area under the ACF before the ACF hits zero for the first time as the time lag increases".

6) Figure 1: Please indicate what was the parcellation used here, and how the time series were averaged within regions. A parcellation with 360 regions was used in other results. However, it is not clear if this was also the case for this comparison between EEG and fMRI.

7) Figure 1C and Figure 1—figure supplement 2. These results show the correlation between the intrinsic timescales obtained from fMRI and from EEG at specific frequency bands. Is the correlation also significant if the EEG signal is considered without filtering at specific frequency bands?

8) Figure 2A: Please clarify the colorbar. How was the intrinsic timescale =1 defined?

9) As shown in Table 2, the regions have various cluster sizes. How was the intrinsic timescales computed in Figure 3? Was it averaged across voxels within the cluster size, or did it have a fixed volume? Please clarify.

Discussion

10) The first sentence proposes an interpretation of the meaning of intrinsic timescales " an index for potentially measuring how long neural information is likely to be stored in each brain region". This is an interesting idea. Unfortunately, however, this interpretation is weak. It has "potentially" and "likely" that make the sentence very apologetic. Moreover, it refers to storing information, but the data corresponds to a resting-state task. Although it is possible to understand the main point, there is still some room for improvements in this definition.

11) "…could potentially be used as a biomarker for early diagnosis of this prevalent neurodevelopmental disorder." This is an intriguing point. Have the authors tested whether it is feasible to propose a biomarker based on the intrinsic timescales in this dataset? What is the specificity and the sensitivity of this measure? If this is not feasible, please explain and discuss the reasons.

12) The gray-matter volume was nicely motivated in the Results section: " We focused on GMV because theoretically, an increase in neuronal density, which is measured by GMV [25], would enhance recurrent neural network activity, and then enlarge the autocorrelation strength in the neural signals". It seems that the manuscript could be improved by incorporating some discussion on this topic.

*Reviewer #3:*

This study used an EEG-validated fMRI measure of intrinsic timescales (IT), along with a structural imaging measure of gray-matter density to compare brain regions in subjects with ASD and typical controls. They find that subjects with ASD show lower IT in the bilateral postcentral gyri and right inferior occipital gyrus, and higher IT in the right caudate. These results correlate both with ASD symptoms, and gray matter density. These findings were replicated in two independent data sets. They also looked at the change in time scales over development in adolescents, and found that these findings correlated with symptomatic progression.

The authors ask well articulated questions and use suitable methodology. While their measure of autocorrelations is innovative, it appears to be well justified by EEG, and by the consistency of their findings. This exploratory case-control study indicates a new direction that can scale to studies in larger and more diverse ASD populations.

My major comment about the work regards the interchangeability in their discussion between timescales observed via single-neuron recordings and timescales observed in fMRI and Ecog. The autocorrelation decay function in their cited electrophysiology papers (refs 10 and 11) last no longer than 800 ms. In their Figure 1A fMRI autocorrelation function, we see the decay last up to 9 seconds. While it is interesting that both methodologies produce brain-region specific differing of autocorrelation functions, they occur on time durations differing by an order of magnitude. By using references to jointly support the investigation of timescales, the authors imply that they represent the same underlying phenomenon. If they really hypothesize that, much more work needs to be done in the Discussion section (and potentially experimentally), to support the claim. If they are agnostic as to the connection between timescales using these different methodologies, that stance also needs to be made explicit.

---

## [Author Response]

Reviewer #1:Atypical intrinsic neural timescale in autism. I previously supplied some feedback to the authors on their original submission. It's a very interesting paper and has been appropriately revised.1) The between group contrast, ASD>TD rests upon measures that are almost certainly used to estimate the severity of ASD: I'm therefore concerned there is some dependence between the test for finding the clusters (ASD>TD) and the regression against ASD severity – not quite double dipping, but surely some lack of complete independence. Please comment/respond to this statistical issue.

We agree on this concern. Four of the eight brain regions of interest (ROIs) found in the ASD-TD contrast did not show significant correlations with ASD severity, but it is still difficult to see complete statistical independence between the whole-brain analysis (Figure 2; Table 2) and the following correlation analysis (Figure 3).

To address this issue, we have now conducted a new correlation analysis with the two independent datasets that had been used in the reproducibility test but not used in the ROI definition.

Note that, unlike the reproducibility test, the ROIs in this additional analysis were defined by the clusters found in the original main analysis (Figure 2, Table 2). Thus, the new association analysis is statistically independent of the whole-brain analysis that searched for the ROIs.

In this new analysis, we found results consistent with our original findings: significant negative correlations between the total ADOS score and the intrinsic neural timescales in the bilateral post central gyri and right IOG (*rho* ≤ –0.60), and a positive association between the ADOS RRB scores and the intrinsic timescale in the right caudate (*F* ≥ 6.0, *P* ≤ 0.03).

These new results were added into the Results section with a new figure (Figure 3—figure supplement 1) as follows:

Results section

“These brain-symptom associations were preserved even when we conducted this association analysis in a more statistically rigorous manner (Figure 3—figure supplement 1). That is, we applied the same ROI sets to two independent fMRI datasets (ETH Zürich and Indiana University datasets; Supplementary Table 1 in Supplementary File) that were not used in the ROI search, and found negative correlations between the intrinsic timescales and the ADOS total scores in the bilateral postcentral gyri and right IOG (*rho* ≤ –0.60) and a significant association between the intrinsic timescale and the ADOS RRB scores in the right caudate (*F* ≥ 6.0, *P* ≤ 0.03 in one-way ANOVAs).”

2) The developmental data set yields very intriguing findings: Are the ROI's used here those discovered in the first cohort? If so, this is important and should be highlighted. Either way, this is a small cohort and should be noted as such.

As the reviewer indicated, the ROIs used in the longitudinal data analysis were defined by the clusters found in the first analysis using the adult data (Table 2). We have now clarified this by adding the following description into the Results and Methods section.

Results section

“We traced the developmental trajectories of the intrinsic neural timescales of the four ROIs whose intrinsic timescales were atypical in the ASD group and associated with the severity of the symptoms. These four ROIs were defined as clusters found in the whole-brain analysis using the adult fMRI data (Figure 2, Table 2).”

Materials and methods section

“We then extracted intrinsic timescales for the eight brain regions of interest (ROIs). The ROIs were defined as clusters found in the main MRI analysis (Table 2), and the neural timescales of the regions were given by the average within the corresponding clusters.”

3) The authors provide nice correlations between fMRI and EEG-based intrinsic time scales (Figure 1): however these differ by two orders of magnitude: This difference should be highlighted in the Discussion and other potential factors, such as neurovascular, should be mentioned.

According to the reviewer’s suggestion, we have now added the following paragraph into the Discussion section.

Discussion section.

“We validated this definition of the intrinsic neural timescales using the simultaneously recorded EEG-fMRI data. […] In fact, when we convolved the EEG signals with the hemodynamic response function (HRF) to take into account such neurovascular coupling, the resultant intrinsic timescales based on HRF-convolved EEG signals were similar in the magnitude to those based on the fMRI data (Figure 1C).”

Reviewer #2:Watanabe, Rees, and Masuda propose to compare the intrinsic timescales of brain regions in ASD and TD. They find significant differences in brain regions that are associated with the severity of symptoms and their gray-matter volume. This is an outstanding contribution to the understanding of structure, dynamics, and function in the human brain. The results are compelling and the analysis is thorough. I naturally recommend that the paper is published. To improve the presentation and the reproducibility of the results the authors should consider some minor suggestions:1) The study finds a variety of regions that have atypical intrinsic timescales and the timescales of some of these regions have significant association with ADOS and with RRB. However, the authors do not discuss whether those regions have been reported before in other ASD studies. This should be discussed, and, if they have not been reported before, the authors might want to highlight the sensitivity of their new proposal.

We appreciate the reviewer’s suggestion. Now we have added the following discussion on this topic into the Discussion section.

Discussion section

“Some human neuroimaging researches examined autistic local neural dynamics in the sensory-related areas and reported observations that are consistent with the current findings. […] However, to the best of our knowledge, no prior research has been conducted on intrinsic neural timescales or signal variability of the caudate in autism.”

Introduction2) The authors mention "functional hierarchies" in the Introduction. Although this is a nice motivation, the concept of "hierarchy of timescales" should also be introduced. The hierarchy of timescales proposes a strong link between the well-established hierarchy in brain structure with a hierarchy in brain function with peripheral regions (at the bottom of the hierarchy) exhibiting fast timescales and core regions (at the top of the hierarchy) exhibiting slow timescales. This concept is proposed by Kiebel et al., 2008, and thoroughly explored by Gollo et al., 2015. Moreover, a distinct functional role of regions at the bottom and top of the hierarchy was demonstrated by Cocchi et al., 2016, using brain stimulation.

We appreciate the reviewer indicating these key studies that we had failed to mention. We have now modified the Introduction by referring to these critical works as follows:

Introduction section

“Computational studies propose that such neural timescales should show a rostrocaudal gradient in the brains [Kiebel, Daunizeau and Friston, 2008] and that densely interconnected central regions, such as prefrontal and parietal areas, should have slower timescales compared to peripheral sensory areas [Chaudhuri et al, 2015; Gollo et al., 2015]. […]The heterogeneity of the neural timescale is considered to be a basis of the functional hierarchy in the brain.”

3) The authors mention "core symptoms" in the Introduction without providing additional details. A brief introduction and motivation of the different (social, communication, RRB) measures is missing in the manuscript, and would improve the presentation. Moreover, it is not clear how ADOS (total) was computed.

We are sorry for our insufficient descriptions in the original manuscript. We have now added clearer explanations as follows:

Regarding “core symptoms” of ASD (Introduction section).

“In fact, the core symptoms of this prevalent neurodevelopmental disorder — challenges in socio-communicational skills and repetitive, restricted behaviours (RRB) — are often linked to atypical information processing”

Regarding the different measures of ASD symptoms (Results section)

“Because previous studies indicate that atypical neural information processing is a common basis for various ASD symptoms, we first examined associations between the neural timescales and the overall severity of this disorder (ADOS total scores). When no significant link was found in this analysis, we then calculated associations between the neural timescales and specific core symptoms.”

Regarding the definition of ADOS total scores (Materials and methods section,)

“The ADOS total scores were defined by the sum of the ADOS social, ADOS communication, and ADOS RRB scores.”

Results/Material and methods4) It is unclear why the results start with a negative sentence explaining what was not done. Instead of this approach, it would be better to provide a brief summary of what the results will cover (the cohorts, EEG, fMRI, structural imaging, symptom measures).

According to the reviewer’s suggestion, we have removed the original sentences starting the Results section, and now added the following brief summary of what the results would cover:

Results section

“First, we introduced a measurement of the intrinsic neural timescales for resting-state fMRI (rsfMRI) signals, and validated it using simultaneous EEG-fMRI data [Deligianni et al, 2014; 2016]. […] Finally, we explored neuroanatomical bases of the intrinsic neural timescales. The reproducibility of our findings was tested using two independent MRI datasets.”

5) "The largest time lag in calculating the initial positive period was set to the value at which the ACF hit zero for the first time as the time lag was increased". This sentence is misleading because it suggests that an interpolation might be required to find when ACF hits zero, and it also contradicts the caption " The initial positive period is the area under the ACF before the ACF hits zero for the first time as the time lag increases".

We are sorry for our confusing description. To be clear, the current method does not require such an interpolation. To clarify this point, we replaced the original sentence with the following one:

Results section

“The upper limit of this period was set at the discrete time lag value just before the one where the ACF became non-positive for the first time.”

6) Figure 1: Please indicate what was the parcellation used here, and how the time series were averaged within regions. A parcellation with 360 regions was used in other results. However, it is not clear if this was also the case for this comparison between EEG and fMRI.

We are sorry for our insufficient descriptions. In fact, this EEG-fMRI comparison did not use pre-defined parcellation systems. Instead, we conducted source reconstruction of the EEG data using DIPFIT2 function in EEGLAB, and identified MNI coordinates for independent components of the EEG data. We then defined regions of interest (ROIs) as 4mm-radius spheres around these MNI coordinates. An intrinsic neural timescale for each ROI was set as the average of the intrinsic timescales within the sphere. To clarify this issue, we added the following statements in the legend for Figure 1.

Legend for Figure 1

“The fMRI-based intrinsic timescale represents the index value averaged over a 4mm-radius sphere whose centre was determined by source reconstruction of independent components of EEG data.”

7) Figure 1C and Figure 1—figure supplement 2. These results show the correlation between the intrinsic timescales obtained from fMRI and from EEG at specific frequency bands. Is the correlation also significant if the EEG signal is considered without filtering at specific frequency bands?

According to the reviewer’s suggestion, we have examined such a correlation with intrinsic timescales based on non-filtered EEG signals, and found a significant one (*R*^2*^ = 0.40 – see bottom right panel, below). We added this finding into Figure 1—figure supplement 2:

8) Figure 2A: Please clarify the colorbar. How was the intrinsic timescale =1 defined?

The intrinsic timescale represents the sum of the autocorrelation function (ACF) values in the initial positive period of the ACF. In addition, to adjust differences in the temporal resolution of the neural data, we multiplied the sum of ACF values by the repetition time (TR) of the fMRI recording, and used the product as an index for the intrinsic timescale. For example, in Figure 1A, the intrinsic timescale = 1 means that the sum of the green bars (ACF values × TR) is equal to 1.

We clarified this by modified the relevant statements in the Results section and legend for Figure 1 and 2 as follows:

Results section

“First, we estimated the sum of autocorrelation function (ACF) values in the initial positive period of the ACF (i.e., the sum of the area of the green bars in Figure 1A). […] This product was used as an index for intrinsic neural timescales.”

Legend for Figure 1

“To estimate an intrinsic neural timescale of an fMRI signal, we first calculated the sum of autocorrelation function (ACF) values of the signals in the initial positive period of the ACF. The period is the area under the ACF up to the time lag value just before the one where the ACF becomes non-positive for the first time as the time lag increases. We then multiplied the obtained area under the ACF by the repetition time (TR), which defined the index for the intrinsic timescale.”

Legend for Figure 2

“The colour bar indicates the intrinsic neural timescales (Figure 1A).”

9) As shown in Table 2, the regions have various cluster sizes. How was the intrinsic timescales computed in Figure 3? Was it averaged across voxels within the cluster size, or did it have a fixed volume? Please clarify.

The intrinsic timescales shown in Figure 3 are the average of the local brain dynamics index within the clusters. To clarified this, we added the following statements into the Materials and methods section and legends for Figure 3.

Materials and methods section

“The regions of interest (ROIs) were defined as clusters found in the whole-brain analysis stated above (Table 2), and an intrinsic timescale for each ROI was given by the average within the corresponding cluster.”

Legend for Figure 3A

“The intrinsic timescales of the regions represent the averages of the local dynamics index within the clusters.”

Discussion10) The first sentence proposes an interpretation of the meaning of intrinsic timescales " an index for potentially measuring how long neural information is likely to be stored in each brain region". This is an interesting idea. Unfortunately, however, this interpretation is weak. It has "potentially" and "likely" that make the sentence very apologetic. Moreover, it refers to storing information, but the data corresponds to a resting-state task. Although it is possible to understand the main point, there is still some room for improvements in this definition.

We agree with the reviewer that, given that it is based on the resting-state data, the original interpretation of the intrinsic neural timescales is not adequate. Therefore, we have now modified the description as follows:

Discussion section

“We investigated the intrinsic neural timescale, whose length is closely related to the functional hierarchy in the brain [Hassan, Chen and Honey, 2015; Murray et al., 2014; Cocchi et al., 2006], in high-functioning individuals with autism.”

11) "…could potentially be used as a biomarker for early diagnosis of this prevalent neurodevelopmental disorder." This is an intriguing point. Have the authors tested whether it is feasible to propose a biomarker based on the intrinsic timescales in this dataset? What is the specificity and the sensitivity of this measure? If this is not feasible, please explain and discuss the reasons.

We admit that the original description sounds an overstatement. In fact, we cannot examine the feasibility of the claim in the current datasets that include no data collected from much younger children. Given this, we have now modified the sentence as follows:

Discussion section

“These findings imply that such atypicality in the temporal characteristics of local neural activity may be one of the basic neuro-aetiologies of autism, which needs to be tested using data collected from much younger children with ASD in future studies.”

12) The gray-matter volume was nicely motivated in the Results section: " We focused on GMV because theoretically, an increase in neuronal density, which is measured by GMV [25], would enhance recurrent neural network activity, and then enlarge the autocorrelation strength in the neural signals". It seems that the manuscript could be improved by incorporating some discussion on this topic.

According to the reviewer’s suggestion, we have now added the following discussion on this issue into the Discussion section.

Discussion section

“The association between the intrinsic timescales and GMVs is theoretically reasonable. […] Although future studies have to directly examine this hypothesis, this logic accounts for the significant correlations between the local neural dynamics and local neuroanatomical structures.”

Reviewer #3:This study used an EEG-validated fMRI measure of intrinsic timescales (IT), along with a structural imaging measure of gray-matter density to compare brain regions in subjects with ASD and typical controls. They find that subjects with ASD show lower IT in the bilateral postcentral gyri and right inferior occipital gyrus, and higher IT in the right caudate. These results correlate both with ASD symptoms, and gray matter density. These findings were replicated in two independent data sets. They also looked at the change in time scales over development in adolescents, and found that these findings correlated with symptomatic progression.The authors ask well articulated questions and use suitable methodology. While their measure of autocorrelations is innovative, it appears to be well justified by EEG, and by the consistency of their findings. This exploratory case-control study indicates a new direction that can scale to studies in larger and more diverse ASD populations.My major comment about the work regards the interchangeability in their discussion between timescales observed via single-neuron recordings and timescales observed in fMRI and Ecog. The autocorrelation decay function in their cited electrophysiology papers (refs 10 and 11) last no longer than 800 ms. In their figure 1A fMRI autocorrelation function, we see the decay last up to 9 seconds. While it is interesting that both methodologies produce brain-region specific differing of autocorrelation functions, they occur on time durations differing by an order of magnitude. By using references to jointly support the investigation of timescales, the authors imply that they represent the same underlying phenomenon. If they really hypothesize that, much more work needs to be done in the Discussion section (and potentially experimentally), to support the claim. If they are agnostic as to the connection between timescales using these different methodologies, that stance also needs to be made explicit.

We have sympathy with the reviewer’s concern. The EEG-based neural timescales were significantly correlated with those based on fMRI data (Figure 1B). In addition, if we convolved hemodynamic response function to the EEG-based data, the neural timescales showed similar values compare to the fMRI-based one (Figure 1C). However, as the reviewer indicated, these observations do not necessarily mean that the same relationship should be found with neural timescales based on spike activity data.

We have explicitly stated this limitation by adding the following paragraph into the Discussion section.

Discussion section

“These EEG-fMRI comparisons indicate that the fMRI-based neural timescales represent an aspect of local neuronal activity. […]To clarify this issue, future studies have to directly compare the fMRI-based neural timescales with those based on neuronal spike activities that are collected simultaneously with fMRI data.”